# Geometric Morphometric Shape Analysis of Mandibular Post-Canine Dentition

Srikant Natarajan [1], Junaid Ahmed [2,*], Shravan Shetty [3], Nidhin Philip Jose [3], Sharada Chowdappa [4] and Kavery Chengappa [5]

[1] Department of Oral Pathology and Microbiology, Manipal College of Dental Sciences Mangalore, Manipal Academy of Higher Education, Manipal 576104, India; srikant.n@manipal.edu
[2] Department of Oral Medicine and Radiology, Manipal College of Dental Sciences Mangalore, Manipal Academy of Higher Education, Manipal 576104, India
[3] Department of Orthodontics and Dentofacial Orthopaedics, Manipal College of Dental Sciences Mangalore, Manipal Academy of Higher Education, Manipal 576104, India; shravan.shetty@manipal.edu (S.S.); nidhin.philip@manipal.edu (N.P.J.)
[4] Independent Researcher, Mangalore 575003, India; mzsharada@rediffmail.com
[5] Department of Public Health Dentistry, Manipal College of Dental Sciences Mangalore, Manipal Academy of Higher Education, Manipal 576104, India; kavery.s@learner.manipal.edu
* Correspondence: junaid.ahmed@manipal.edu

**Abstract: Background**: Genetic and epigenetic alterations have significant impacts on the morphology of permanent mandibular premolars and molars. Geometric morphometry is a powerful technique, which can be utilized to identify specific landmarks that exhibit variation and that are associated with ancestry and dimorphism. **Methods**: The geometric and anatomic landmarks of mandibular premolars and molars were extracted from 3D digital replicas of diagnostic dental casts prepared for model analysis (n = 160). Tooth shape analysis was conducted using various techniques, including Procrustes superimposition, Procrustes ANOVA, discriminant function analysis, and the regression of shape over the centroid size. **Results**: Procrustes ANOVA showed that centroid size was not significantly different between the two sexes, but shape was significantly different in the two-cusp-type second premolars ($p = 0.0035$) and in the first/second molars ($p < 0.001$). The three-cusp type of the second premolars showed the highest degree of allometry, with 3.35%, followed by the mandibular second molars, with 3%, indicating that distal class types have a tendency to exhibit allometry. The distal and lingual components of the tooth showed more variability, and females tended to have sharper cusp configurations. **Conclusions**: This study shows how landmarks vary in permanent human post-canine dentition, a crucial finding for anatomic reconstruction and restorative dentistry. In particular, the molars and premolars of the mandible post-canine teeth are critical for achieving optimal masticatory efficiency and overall health. Additionally, a higher degree of allometry and the later formation of cusps correlate with greater shape variation, particularly on the distal and lingual sides. For precise restorative procedures, a thorough understanding of the anatomy of these teeth is required.

**Keywords:** geometric morphometry; sexual dimorphism; landmark; Procrustes ANOVA; 3D; human variation; enamel thickness; dentistry; tooth

## 1. Introduction

The shape of teeth is a fascinating subject that is influenced by a variety of factors. Genetics, environment, and epigenetics all play roles in determining the shapes of an animal's teeth [1]. From a genetic perspective, the shapes of an animal's teeth can reveal its genealogical relationship to other animals. Environmental factors, such as diet, can also have significant impacts on tooth form. Finally, epigenetics can contribute to variations in tooth shape within a species, or even within a single individual [1]. Understanding

the factors that influence tooth form is important for a variety of reasons. It can help us to better understand the evolution of different species, and how they are related to one another. It can also help us to understand the impact of diet on tooth health, and how we can improve our own dental health. By studying tooth form, we can gain a deeper understanding of the complex interplay among genetics, environment, and epigenetics in shaping the natural world.

Quantifying and analyzing the shapes of morphological structures has long been a challenge for researchers. However, new methods have been developed to tackle this issue, including techniques for acquiring outline and landmark data. Outline methods involve digitizing points along the outline of a structure and fitting them using mathematical functions [2]. On the other hand, landmark-based geometric morphometric methods can be applied to the 2D or 3D coordinates of biologically defined landmarks [3–5]. Procrustes analysis is a popular technique that superimposes landmark configurations using least-squares estimates for translational and rotational parameters. In recent years, procrustes analysis has been used to analyze the buccal and occlusal shapes of human teeth, resulting in the illustration of the mean shape of teeth [6,7]. These landmark-based methods provide a deeper understanding of shape variations within a species and offer insights into the evolution of phenotypes. Overall, these new methods have revolutionized the way researchers analyze morphological shape, allowing for more accurate and detailed studies of biological structures [1,8].

The study of human dentition is a complex and fascinating field that has garnered significant attention over the years. Researchers have conducted numerous investigations to determine the mode of inheritance for various dental characteristics, using metric analyses by measuring tooth dimensions (odontometry), ratios (tooth indices), as well as non-metric dental traits. Despite extensive research, there are still differing opinions on the role of linked genes in the formation of tooth size and shape. While some studies have suggested that X-linked genes play a role in size development [9,10], others have demonstrated that a linkage model is not necessary to explain the observed dimorphism [11–13].

The purpose of this study was to investigate the relationship between tooth shape and sex. By examining the dental casts of individuals from the Dakshina Kannada region, we hoped to gain insight into the variation in shape of the mandibular post-canine dentition. This research has the potential to identify landmark variations that would define morphology for tooth restoration and human identification in this region and beyond. The aim of this study was to examine the changes in geometric and anatomic landmarks and, thus, the three-dimensional shape of the mandibular post-canine dentition, using 3D geometric morphometric analysis. Further, we aimed to determine the specific landmarks in post-canine dentition that show sexual dimorphism.

## 2. Materials and Methods

*Sample Size Calculation*

In a study conducted by Yong R et al. (2018), [8] it was found that there is a difference in the shape of human premolars between males and females. Specifically, the centroid sizes of the upper premolars were significantly larger in males (first premolar: 28.33 ± 1.96; second premolar: 27.54 ± 1.83) compared to those in females (first premolar: 27.82 ± 1.93; second premolar: 26.93 ± 2.05) [8]. The sample size was determined using Equation (1) to compare means of the two groups.

The equation used was

$$N = \frac{2\left(Z_{1-\frac{\alpha}{2}} + Z_{1-\beta}\right)^2 \sigma^2}{d^2} \tag{1}$$

where N is the sample size, *d* is the clinically significant difference, and $\sigma$ is the mean standard deviation. After considering the centroid values of males and females in the upper first and second premolars, with a 5% alpha error (z = 1.959964) and 80% power

(z = 0.841621), taking a clinically significant difference of 1 unit, the required sample size for each sex was calculated to be 60 individuals.

Prospective pretreatment dental casts (n = 160) were obtained from the archives of the Department of Orthodontics at the Manipal College of Dental Sciences, Mangalore, as per the sample size calculation mentioned above. The sample size determined was 120 individuals; but, in order to account for anatomic variation in the mandibular second premolar, an additional 40 casts were taken, for examination of the 60 male and 60 female casts of all post-canine teeth. This study was undertaken following clearance from the institutional ethics committee. Study casts of patients in the age range of 13–20 years were included in this study. After the age of 20, when the individual's occlusion has developed, the contacts and cusp tips tend to show attrition, leading to the development of proximal contact areas and occlusal wear facets. This can result in incorrect landmark acquisition, particularly in terms of proximal contacts and cusp tips. To prevent such errors, our sample was drawn from the age range of 13–20 years, where the landmarks are stable, and no physiological changes are observed so early in life. Comprehensive demographic history was recorded, including the birthplace and domicile of each individual, as well as those of their parents and grandparents. Individuals were included in this study if they, along with their parents and grandparents, were born and raised in the Dakshina Kannada region. This three-generation residency criterion was established to ensure a strong regional connection among participants. Verification of this information was conducted through patient interviews at the dental clinic, where questions about the birthplaces, residences, and occupations of their parents and grandparents were asked. In order to avoid confounding factors and errors in evaluation, the inclusion and exclusion criteria listed in Table 1 were closely adhered to.

**Table 1.** Inclusion and exclusion criteria.

| Inclusion Criterion | Exclusion Criterion |
| --- | --- |
| • Individuals from Dakshina Kannada, between 13 and 20 years of age, were included in this study.<br>• Individuals having a full complement of posterior teeth from the first premolar to the second molar in both sides were included in the study | • Any tooth in the posterior segment that has undergone restoration, or has fractures, caries, or crowns, was excluded from this study.<br>• Patients exhibiting any developmental anomalies that affect the shape or size of the tooth, as well as individuals displaying symptoms of a syndrome, were excluded from this study. |

After obtaining the impressions (taken using Dentsply Aquasil Soft Putty and Kit, Dentsply, Noida, India) and pouring the casts using Type 4 Extra Hard Dental Die Stone (Zhermack Badia Polesine (RO), Italy), we promptly digitized them using the advanced inEOS X5-Lab scanner (Dentsply Sirona, Noida, India). To accurately define landmarks, we identified key landmarks based on both anatomic and geometric evidence. Our approach was substantiated by the seminal work of Biggerstaff R (1969), who proposed a method for describing the basal area of the posterior teeth [14]. We also drew on the landmark definitions suggested by Robinson DL et al. (2002) and Al-Shahrani I et al. (2014), modifying them as needed to suit our specific needs [5,15].

Landmarks were identified based on "Anatomic evidence" (corresponding to cusp tips, fissure junctions, and the endpoints of common clinical measurements like mesiodistal width, buccolingual width, line angles, and point angles) and "Geometric evidence" (based on the crests of curvatures, line and point angles, and surface landmarks corresponding to occlusal surface landmarks). These landmarks were marked on the 3D models using the open-source 3D Slicer software, version 4.10.2 (http://www.slicer.org) accessed on 25 December 2023 [16]. The number of landmarks varied according to the complexity of the

tooth involved. The first premolar, the two-cusp-type second premolar, the three-cusp-type second premolar, the first molar, and the second molar had totals of 20, 19, 21, 32, and 27 landmarks, respectively. Of these landmarks, 11, 11, 13, 19, and 15 were based on geometric evidence.

In total, 120 dental casts with the presence of all 8 of the relevant teeth, i.e., the first premolar to the second molar, bilaterally, were selected for shape analysis. These 120 casts consisted of those from 60 males and 60 females, and were specifically analyzed in relation to the mandibular first premolar, mandibular first molar, and mandibular second molar. In the sample of 120 patients, it was found that 37 had the two-cusp type of mandibular second premolar, while 83 had the three-cusp type. In order to increase the accuracy of our analysis, we expanded our sample size to include 40 additional casts (making the total sample number n = 160). This allowed us to examine the shape variations in both the two-cusp (n = 49) and three-cusp (n = 111) types of mandibular second premolars (Table 2).

**Table 2.** Demographic details and the distribution of patients by sex.

| Parameter | Details | |
|---|---|---|
| Age | | |
| Total (Mean ± Standard Deviation) | 18.62 ± 2.49 | |
| Female (Mean ± Standard Deviation) | 17.85 ± 2.81 | |
| Male (Mean ± Standard Deviation) | 19.47 ± 1.74 | |
| Sex | | |
| Male (n,%) | 76 (47.5%) | |
| Female (n,%) | 84 (52.5%) | |
| Distribution of cases | | |
| Tooth | Male | Female |
| Mandibular First Premolar | 60 | 60 |
| Mandibular Second Premolar (Two-cusp Type) | 29 | 20 |
| Mandibular Second Premolar (Three-cusp Type) | 47 | 64 |
| Mandibular First Molar | 60 | 60 |
| Mandibular Second Molar | 60 | 60 |
| Number of Landmarks | | |
| Tooth | Geometric Evidence | Anatomic Evidence |
| Mandibular First Premolar | 11 | 9 |
| Mandibular Second Premolar (Two-cusp Type) | 11 | 8 |
| Mandibular Second Premolar (Three-cusp Type) | 13 | 8 |
| Mandibular First Molar | 19 | 13 |
| Mandibular Second Molar | 15 | 12 |

Using a random subsample of 20 dental models, the assessment of intra- and interobserver error in landmark location was carried out. Over the course of five days, landmarks were marked independently by two observers. A single investigator obtained the landmarks to study for intra-observer variation. Reliability was measured using the Interclass Correlation Coefficient (ICC) [17] (IBM, SPSS version 20.0, Chicago, IL, USA), which ranged from 0.982 to 1.000, showing excellent reliability. Intra-operator reproducibility at the digitization stage was assessed using a one–way nonparametric ANOVA, with a randomized permutation procedure (10,000 iterations), using MorphoJ software. The average variance related to digitization error for centroid size and shape among the selected teeth

was <10%, leading to the conclusion that the errors were negligible compared to the total measurement effect.

The landmark coordinates were tabulated and processed in Microsoft Excel (Microsoft Corporation, 2023) and imported to the morphometry software MorphoJ (version 1.08.01, for Microsoft Windows, Klingenberg, C. P. 2011, https://morphometrics.uk/MorphoJ_page.html, accessed on 25 December 2023), which is an open-source integrated program package for performing geometric morphometrics and statistics for the same [3]. To evaluate the landmarks involved in shape variation, landmarks were normalized using generalized Procrustes analysis, followed by principal component analysis. Discriminant function analysis was then performed to assess the shape differences demonstrated by landmark coordinates, contributing to the observed dimorphism. Additionally, allometry, or size-dependent shape difference, was evaluated using a regression analysis of the shape coordinates and centroid size. A 95% confidence interval was used to determine the level of significance. A *p*-value of less than or equal to 0.05 was considered statistically significant.

### 3. Results

Our study comprised 160 patient casts from 76 males and 84 females, with a mean age of 18.62 ± 2.49 years (Table 2). Principal component analysis showed that the number of components, explaining 80% of the variation in the shapes of the mandibular first premolar, the two-cusp-type second premolar, the three-cusp-type second premolar, the first molar, and the second molar, were 15/53 (28.3%), 14/50 (28%), 15/56 (26.79%), 21/89 (23.6%), and 18/84 (21.43%) respectively. Procrustes superimposition showed a uniform scatter of landmarks round the median landmark in both the xy- and xz-axes.

In the mandibular first premolar, the scatter plot of the principal components (PCs) 1 and 2 (PC1 and PC2) showed homogenous distribution (Figure 1) with the graphs, showing maximum variations in the mesiobuccal and distopalatal ends of the tooth outline, as well as in the mesiolingual groove ending (Figure 2). Analysis of shape and centroid size using Procrustes ANOVA showed that centroid sizes were not significantly different between the sexes ($p > 0.05$) in all teeth (Table 3). The shape of the tooth, however, was significantly different between the sexes in the two-cusp-type mandibular second premolar ($p = 0.0035$), mandibular first molar ($p < 0.0001$), and mandibular second molar ($p < 0.0001$). Regression analysis showed that size-dependent shape variation was noted in only 1.38% of cases, indicating no significant allometry ($p = 0.56$) (Figure 3, Table 4). Discriminant function analysis also showed that there was 80.833% accuracy in distinguishing sexes, but this was not significant, with a *p* value of 0.3181 (Table 5).

**Table 3.** Procrustes ANOVA to detect shape differences between sexes.

| | Mandibular First Premolar | | Two-Cusp-Type Mandibular II Premolar | | Three-Cusp-Type Mandibular II Premolar | | Mandibular First Molar | | Mandibular Second Molar | |
|---|---|---|---|---|---|---|---|---|---|---|
| | **Centroid** | **Shape** | **Centroid** | **Shape** | **Centroid** | **Shape** | **Centroid** | **Shape** | **Centroid** | **Shape** |
| Sum of Squares | 0.01141 | 0.0334655 | 0.084 | 0.0263 | 0.03839 | 0.0135 | 1.208 | 0.01269 | 5.8572 | 0.01891 |
| Mean Sum of Squares | 0.01141 | 0.00063 | 0.084 | 0.00052 | 0.03839 | 0.00024 | 1.208 | 0.00014 | 5.8572 | 0.00025 |
| F | 0.02 | 1.17 | 0.09 | 1.63 | 0.05 | 0.95 | 0.78 | 1.71 | 3.01 | 1.95 |
| *p* value | 0.9026 | 0.156 | 0.7717 | **<u>0.0035</u>** | 0.8312 | 0.5989 | 0.3780 | **<u><0.0001</u>** | 0.0853 | **<u><0.0001</u>** |

Bold and underlined numbers represent statistically significant association.

**Table 4.** Regression of the shape parameters with the centroid size of the tooth.

| | Mandibular First Premolar | Two-Cusp-Type Mandibular Second Premolar | Three-Cusp-Type Mandibular Second Premolar | Mandibular First Molar | Mandibular Second Molar |
|---|---|---|---|---|---|
| Total SS: | 2.308619 | 0.78446 | 1.5874 | 0.891322 | 1.162052 |
| Predicted SS: | 0.031928 | 0.02264 | 0.05311 | 0.020819 | 0.034882 |
| Residual SS: | 2.276692 | 0.76182 | 1.5343 | 0.870504 | 1.127169 |
| % predicted: | 1.38% | 2.89% | **3.35%** | **2.34%** | **3.00%** |
| *p* value | 0.056 | 0.1405 | **<0.001** | **<0.001** | **<0.001** |

Bold and underlined numbers represent statistically significant association.

**Table 5.** Discriminant function analysis.

| | Permanent Mandibular I Premolar | Mandibular Second Premolar, Two-Cusp Type | Mandibular Second Premolar, Three-Cusp Type | Mandibular First Molar | Mandibular Second Molar |
|---|---|---|---|---|---|
| Procrustes distance | 0.02361 | 0.0472 | 0.0224 | 0.0205 | 0.0251 |
| Mahalanobis distance | 1.8882 | 6.8036 | 2.7051 | 3.7158 | 3.0236 |
| T-square | 106.96 | 547.91 | 198.30 | 414.21 | 274.27 |
| *p* value | 0.3181 | 0.94 | **0.019** | 0.3073 | 0.1063 |
| Prediction Accuracy, Female | 83.333 | 95 | 90.625 | 95.00 | 95.00 |
| Prediction Accuracy, Male | 78.333 | 100 | 91.49 | 95.00 | 88.33 |
| Overall Accuracy | 80.833 | 97.95 | 90.99 | 95.00 | 91.66 |

Bold and underlined numbers represent statistically significant association.

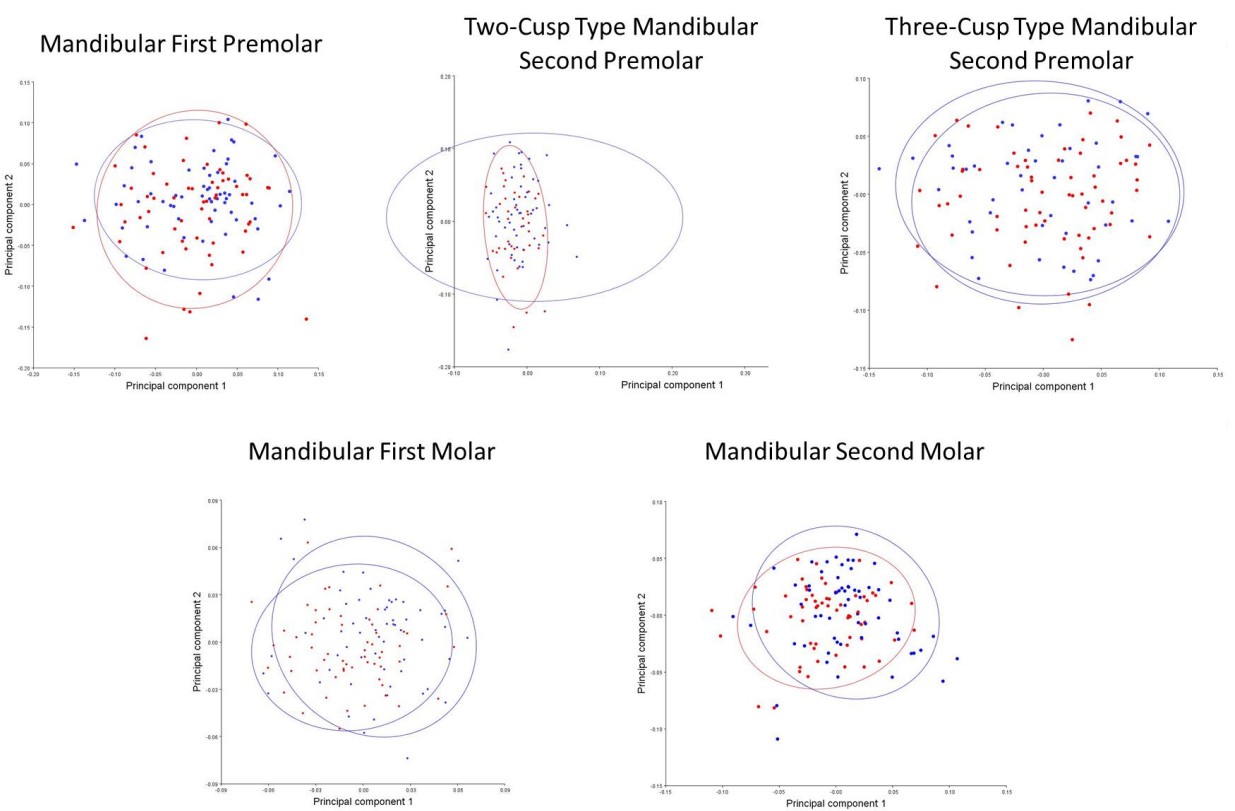

**Figure 1.** Principle component analysis for landmarks of the five types of mandibular post-canine dentition (red dots represent females and blue dots represent males).

## Mandibular Left First Premolar

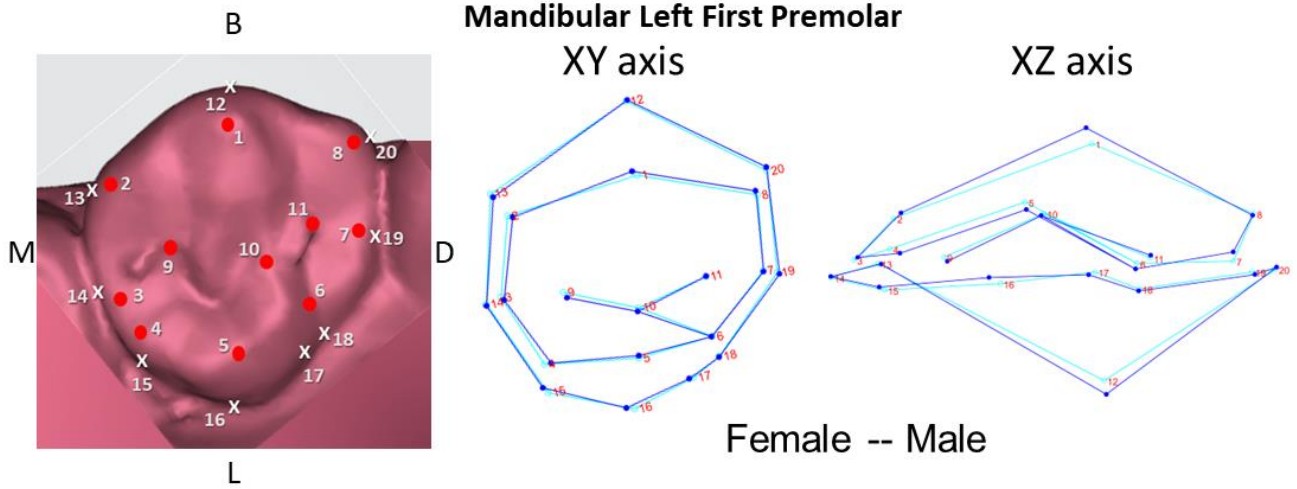

## Mandibular Second Premolar Two-Cusp Type

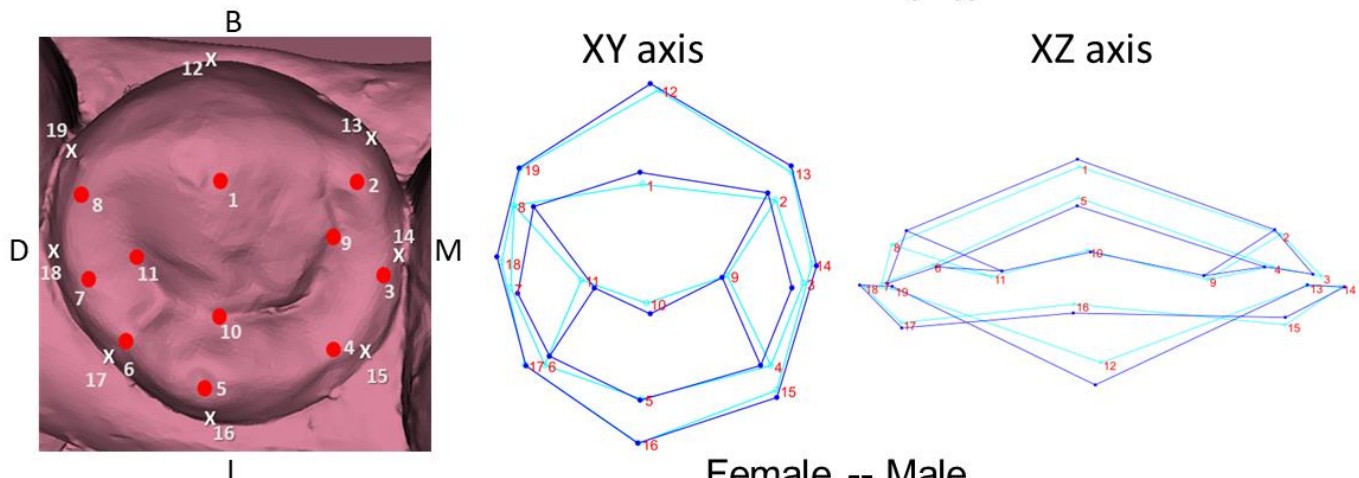

## Mandibular Second Premolar Three-Cusp type

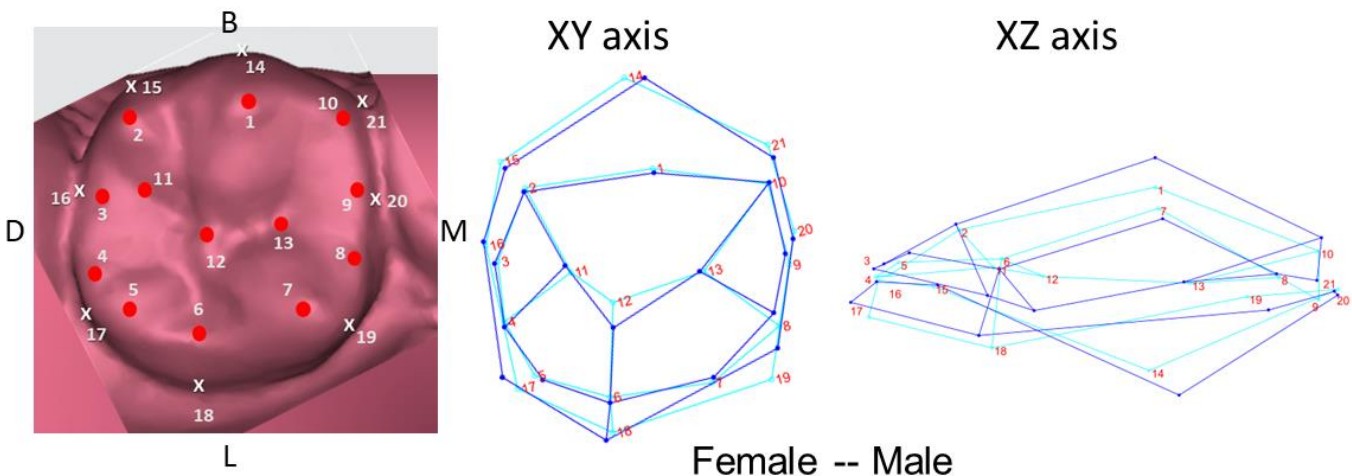

**Figure 2.** Line graphs depicting landmark changes between sexes in the mandibular first and second premolars. Note: dark blue represents female and light blue represents male and the numbers represent the landmarks.

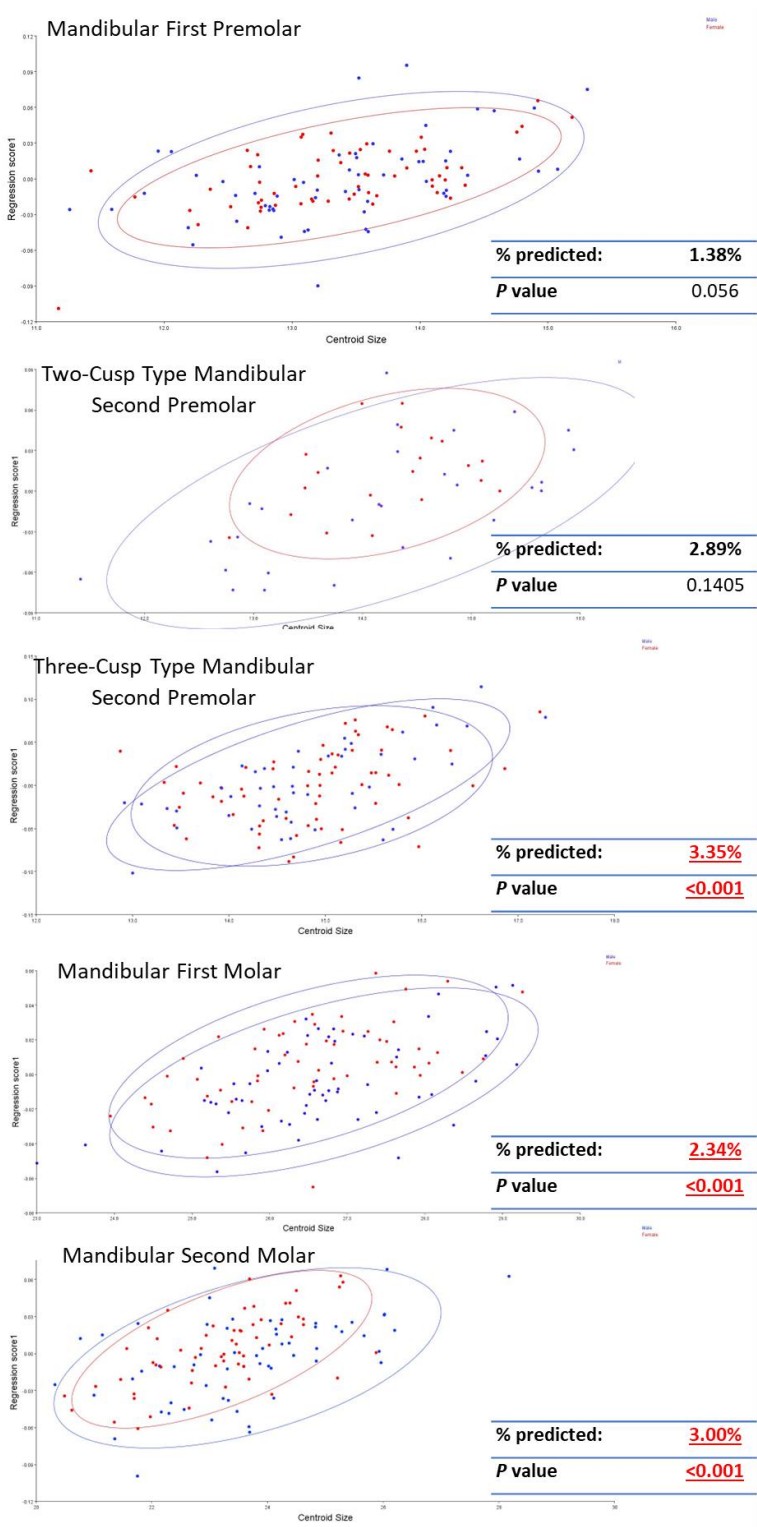

**Figure 3.** Regression analysis showing size-dependent shape variation (allometry). The red dots represent females and blue dots represent males.

The mandibular II premolar showed two morphological subtypes. Among the 160 samples analyzed, 49 demonstrated two-cusp-type and 111 demonstrated three-cusp-type variants. The two-cusp type was represented by 19 landmarks (11 of anatomic evidence, and 8 of geometric evidence) and the three-cusp type was represented by 21 landmarks (13 of anatomic evidence, and 8 of geometric evidence). Procrustes ANOVA showed that centroid size was not significantly different in either type between the sexes. However, the

shape of the two-cusp type was significantly different between the two sexes ($p = 0.0035$), but not that of the three-cusp type ($p = 0.5989$) (Table 3). Discriminant function analysis showed that the shape of the three-cusp-type mandibular second premolar could be used to correctly classify the sexes in 90.99% of the cases ($p = 0.019$). The two-cusp type had higher accuracy, with 97.95%; however, this was not significant ($p = 0.94$) (Table 5). Regression analysis showed that size-dependent shape variation was seen in 2.89% ($p = 0.1405$) of two-cusp-type cases and 3.35% ($p < 0.001$) of the three-cusp-type cases (Figure 3, Table 4).

Procrustes ANOVA, for both the mandibular first and second molars, showed that the centroid size was not significantly different between the sexes. Procrustes ANOVA of the shape of the tooth showed significant differences between the sexes in both the first and second molars, with a $p$ value $< 0.0001$ (Table 3).

The results of the discriminant function analysis revealed that the morphologies of the mandibular first molar and mandibular second molar accurately distinguished between male and female individuals in 95% ($p = 0.3073$) and 91.66% ($p = 0.1063$) of cases, respectively (Table 5). Regression of the centroid size to shape showed that allometry was significant in both the first and second molars, accounting for 2.34% and 3.00%, respectively (Figure 3, Table 4).

In the mandibular first premolars, males showed a narrower distal marginal ridge (landmarks 2, 4, 13, and 14) and a buccally placed central groove (landmarks 9 and 10), as observed in the xy-axis. In the z-axis, which refers to the vertical dimension of a tooth, landmark 1 (representing the buccal cusp) and landmark 12 (representing the buccal crest of curvature) are important indicators of tooth structure. Specifically, the buccal cusp height is higher and the buccal crest of curvature is lower in females, as compared to males. There was no significant variation observed in the mesiolingual groove, which is a unique anatomic feature of this tooth (landmarks 6 and 18) (Figure 2).

The landmarks of the two-cusp type of mandibular second premolar showed that males have a tendency to have narrower marginal ridges (landmarks 7, 8, 18, and 19) and a narrower occlusal table (lingually placed landmark 1). This tooth in the female set had a propensity towards a lingually placed central pit (landmark 10). Examination of the landmarks in the z-axis showed that females have a taller buccal cusp (landmark 1) and shorter lingual cusp (landmark 5) and tooth is wider in males (landmarks 2, 3, 7, and 8) (Figure 2).

Examination of the three-cusp type of mandibular second premolar showed that male teeth are wider than female teeth in relation to the mesiolingual (landmark 19) and distobuccal (landmarks 14 and 15) boundaries. Females have a more lingually placed central pit (landmark 12). The distolingual cusp is larger in females compared to males (quadrangle formed by landmarks 4, 11, 12, and 17) (Figure 2).

Analysis of the wireframe plots of the mandibular first molar showed that the occlusal table is narrower in females than in males, as shown by the lingually placed landmarks 1–5 and buccally placed landmarks 9–11. The central pit is more lingually placed in males (landmark 17). In the z-axis, the level of the mesiobuccal groove between the two buccal cusps is more occlusally placed in males (landmark 2). The cusp tips (landmarks 1, 3, 5, 9, and 11) are placed higher in females compared to males, indicating sharper cusp configurations (Figure 4).

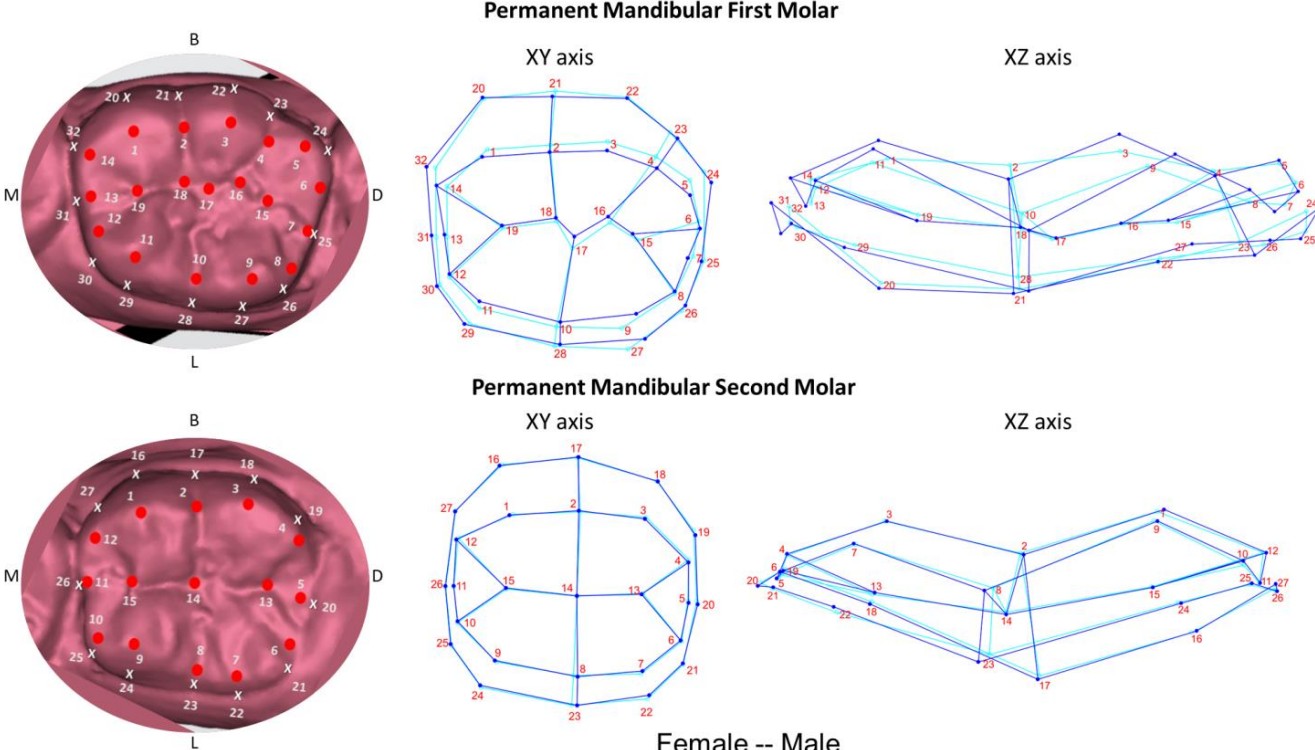

**Figure 4.** Line graphs depicting landmark changes between sexes in the mandibular first and second molars Note:dark blue represents female and light blue represents male and the numbers represent the landmarks.

## 4. Discussion

Tooth morphology is a fascinating and complex subject that is essential for dental professionals to understand. By analyzing the landmarks of dental anatomy, we can gain a better understanding of tooth morphology and improve treatment outcomes. The shape variation in teeth also correlates with ancestry, making it an important area of study for dental professionals. With a thorough understanding of the shape variation in and cusp configuration of the mandibular post-canine dentition, dental professionals can contribute to various fields of work. The present study aims to investigate the shape variations in mandibular premolars and molars, which are crucial for dental treatments such as orthodontics, restorative dentistry, and prosthodontics. Furthermore, documenting population-specific morphological alterations can serve as a reliable indicator of ancestry, providing valuable information for forensic investigations and anthropological studies. This information can help dental professionals identify individuals and provide insight into their ancestry, which can be useful in a variety of settings.

In our study, we observed that the mandibular molars and the two-cusp-type mandibular second premolars displayed dimorphism. This means that there were noticeable differences in the size, shape, and structure of these teeth between males and females. It is possible that sex-linked genes influence enamel and dentin contributions to crown size. Schwartz and Dean (2005) reported that males and females have similar amounts of enamel. However, males have a significantly higher proportion of dentin in the lower molars [10]. The study of individuals with aneuploid sex chromosomes has shown that the X chromosome is associated with enamel thickness, and the Y chromosome is associated with both enamel and dentin thickness [18,19]. The role of hormones in contributing to dimorphism is less likely for the mandibular first and second molars. This is because the crown of the mandibular first molar is completed by the age of 2–3 years, while the crown of the mandibular second molar is completed by the age of 7–8 years. The growth spurt only occurs during the ages of 12–15 years, when the third molar is forming. Therefore, dimorphism associated with

hormonal influence is more likely for the third molar, but not the teeth mesial to it [10,20]. In a recent study, conducted by Monson T et al. (2020), the shape of the enamel–dentinal junction (EDJ) was examined and found to have a significant correlation with genetic drift, rather than adaptive processes, as seen in bone shape. The researchers also emphasized that the shape of the EDJ is a heritable factor, indicating that genetics play a larger role in tooth shape than hormones, habits, or other external factors [21].

Saunders et al. (2007) conducted a morphometric analysis of mandibular premolars and discovered that female teeth had a significantly higher proportion of enamel formed in the cusp area. Specifically, they found that the average enamel cap area was 11.8% greater in females than in males [22]. Our study builds upon this research by revealing that female cusps are positioned higher and have sharper configurations [10]. Further investigation is needed to explore the relationship between the X chromosome and the higher quantity of enamel formation observed in females.

In our study, we observed that the occlusal table of the mandibular first molar was narrower in females. This finding is consistent with the study conducted by Yoo H et al. (2014), who reported that the occlusal table area was significantly smaller in females (44.92 mm$^2$) compared to males (49.90 mm$^2$). However, it is interesting to note that the proportion of the occlusal table to the total tooth area was statistically similar and marginally higher in females (57.69%) than in males (56.11%) [23]. This suggests that, despite having a narrower occlusal table, molars in females exhibit the same or an even greater proportion, indicating no compromise in masticatory efficiency. These findings have important implications for dental professionals and researchers, as they highlight the need to consider gender differences when evaluating dental health and treatment outcomes. Yoo et al.'s study also found that there was no sexual dimorphism of size seen in the second molar, which is similar to the results of our study, in which the wireframes show similar outlines in males and females [23]. The relatively stable second molar formation may be due to its formation at 7–8 years of age, when there are no birth-related changes or puberty-related growth spurts seen to influence its formation [20].

In 2014, Morita and colleagues conducted a study that revealed that the hypocone (distolingual cusp of the maxillary molars) exhibits a greater degree of variation in both size and shape. Interestingly, they also discovered that the later the formation of the cusp, the more variability the tooth will exhibit [24]. This finding aligns with our own research, which showed that the distolingual cusp of the mandibular second premolar (three-cusp type), displays a higher degree of variation. In general, there is a greater degree of variation on the lingual side of the crown compared to the buccal side. This phenomenon can be attributed to several factors, including the mandible's growth pattern, which tends to favor the lingual side, the presence of permanent tooth germs on the lingual side, and the spatial relationship of other osteogenic tissues [24,25].

A study of the hominin premolars (a primate of a taxonomic tribe; Hominini, which comprises those species regarded as human, and those directly ancestral to humans) by Martinon-Torres et al. (2006), showed that the primitive pattern of the premolars was predominantly rectangular, with a larger occlusal polygon and a mesially placed metaconid (lingual cusp). Seventy percent of modern homo sapiens demonstrated a circular outline with a centrally placed metaconid [26]. Both the mandibular first premolar and the two-cusp-type mandibular second premolar showed a homogenous round-to-oval configuration in our sample. Our sample also showed predominance of the three-cusp type (111/160 samples), which may represent the primitive forms of the premolar, or may be due to the overlap of the genetic influence of tooth development of the molar and premolar tooth germs.

In 2004, Hlusko and colleagues conducted a study on the dimorphism of enamel thickness in mammalian molar teeth. Their findings revealed no consistent pattern of dimorphism in absolute enamel thickness. However, when the linear size of the tooth was normalized with the thickness of enamel, the factor was dimorphic, with females having thicker enamel [12]. In our study, we utilized discriminant function analysis to determine

the statistically significant classification of the mandibular second molar with three cusps ($p = 0.019$). This tooth exhibited the highest degree of allometry among the mandibular post-canine dentition, accounting for 3.35%. These results suggest that tooth size is a confounding factor for the dimorphism exhibited by the shape of the tooth. Overall, our study builds upon the previous research conducted by Hlusko and colleagues, providing further insight into the complex relationship between tooth size and enamel thickness for dimorphism. These findings have important implications for understanding the evolution and development of mammalian teeth. Allometry is a crucial factor in morphology, as it plays a significant role in integrating various aspects of an organism's physical structure. Specifically, allometry helps to establish the adult proportions of an organism by integrating body size with the relative size of the viscerocranium. This correspondence is essential for ensuring optimal masticatory efficiency with evolution [27].

The evaluation of the shape of a tooth provides us with valuable insights into the genetic and evolutionary bases of development. Through geomorphometric analysis, we can identify the landmarks that differ between sexes. Knowledge of the relative cusp heights, locations of central pits, and greater variation in the distal and lingual portions of the tooth in relation to the later developmental sequence is crucial for dentists to effectively restore teeth. Standardizing landmarks can prove to be beneficial in creating a uniform set of shape properties for the 3D visualization and reconstruction of tooth anatomy. This process ensures that the data collected are consistent and reliable, allowing for accurate analysis and interpretation. By establishing a standard set of landmarks, researchers and professionals can effectively communicate and compare their findings, ultimately advancing the field of dental research.

Mahn E et al. (2018) correlated the anterior tooth form and gender and found no significant difference between them. They proposed five new hybrid tooth forms, which can be prefabricated for patient use in esthetics [28]. Using our study of the posterior tooth landmarks, clinically and functionally superior preformed crowns, can be fabricated specifically for premolars and molars. Similar to the study by Krenn V et al. (2019) we found that the mandibular premolars vary more on the lingual side than on the buccal segment. They also found that the two mandibular premolars covary with each other [29]. Thus, the landmark data of available teeth can be further assessed to predict the shapes of missing teeth, which can be useful to reproduce the crown clinically.

Symmetry in the dental arches is clinically relevant. The presence of directional asymmetry, i.e., consistently larger or smaller teeth on one side, may be a useful feature for the practice of orthodontics and alignment [30].

Understanding the landmarks can also be useful for anthropological and forensic studies. The landmark changes from early humans to modern humans can provide good insights into the evolution of tooth shape [17,31].

Relying solely on molars for dental profiling has its limitations. While they are important teeth in the dentition, molars are more prone to caries and tooth wear than other teeth. This increased risk can be attributed to their early time of emergence in the oral cavity, which exposes them to bacterial load and masticatory function at an earlier stage. Additionally, molars are more difficult to clean thoroughly, which can lead to destructive processes that alter crown morphology and increase the likelihood of tooth loss [32]. In this regard, our study can be useful for identifying the 3D crown morphology that needs to be reconstructed; however, it may prove less efficient for dental profiling based on shape in anthropological studies.

Moreover, the present report tested casts of patients in the age range of 13–20. It would be interesting, in the future, to test varied age groups, also carefully considering the effects of remineralizing agents, such as fluoride [33], casein phosphopeptide-amorphous calcium phosphate [34], and biomimetic hydroxyapatite [35], in order to evaluate the interaction between age and preventive treatments.

One limitation of this study is its unicentric nature. The geographic overlap of the area served by the institute for oral health care may result in a mixed population with

overlapping genetic pools. Care was taken through interviews to include patients from the Dakshina Kannada region in our study. To enhance the accuracy and validity of the findings, a multicentric study should be conducted. A further limitation of this study is the analysis of the genetic pool. This research correlates the genetic basis of tooth morphogenesis with the shape of the tooth. To truly understand the hypothesis within the given population, further research testing the association between genetics and tooth shape is necessary.

## 5. Conclusions

In conclusion, the mandibular post-canine teeth, specifically the premolars and molars, are crucial for achieving optimal masticatory efficiency and overall health. Their precise restoration requires a thorough understanding of the shape of these teeth. Furthermore, analyzing the shapes of these teeth can aid in dental profiling, which is useful for forensic investigation and anthropology. It is imperative to have a comprehensive understanding of the importance of these teeth in order to maintain proper oral health, and to utilize them effectively in various fields of study.

**Author Contributions:** Conceptualization, S.N. and J.A.; methodology, S.N., S.C. and K.C.; software, S.N. and S.S.; validation, S.N., N.P.J. and S.C.; formal analysis, S.N., S.C. and K.C.; investigation, S.N. and J.A.; resources, N.P.J. and S.S.; data curation, S.N. and S.C.; writing—original draft preparation, S.N.; writing—review and editing, S.N., N.P.J., S.S. and J.A.; visualization, S.N.; supervision, J.A.; project administration, J.A.; funding acquisition, S.N. and J.A. All authors have read and agreed to the published version of the manuscript.

**Funding:** This research was funded by the Science and Engineering Research Board (SERB), Department of Science and Technology, Government of India (file number CRG/2020/001057).

**Institutional Review Board Statement:** This study was conducted according to the guidelines of the Declaration of Helsinki and approved by the Institutional Ethics Committee of the Manipal College of Dental Sciences, Mangalore (protocol code 20018, dated 16 August 2020).

**Informed Consent Statement:** Patient consent was not required for this study, as it utilized the dental study models created for their treatments. These casts were anonymized by the clinician to ensure that the researchers could not identify the individuals involved.

**Data Availability Statement:** The data for this study are available by contacting the corresponding author, J.A. The data are not publicly available due to privacy.

**Conflicts of Interest:** The authors declare no conflicts of interest. The funders had no role in the design of this study; in the collection, analyses, or interpretation of data; in the writing of the manuscript; or in the decision to publish the results.

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
