# Peer review of "Geometric Morphometric Shape Analysis of Mandibular Post-Canine Dentition"

_applsci, doi:10.3390/app14020658_

Round 1

Reviewer 1 Report

Comments and Suggestions for Authors

I congratulate the authors for a well designed and executed research study. I would like to point out a few suggestions and improvements.

1. Can the authors include the inclusion and exclusion criteria in a tabulated form with detailed criteriae.

2. How many operators were involved in the study and how did the authors take care of operator bias.

3. Since this study was done in Dakshina Kannada region, how certain are the authors that the findings of the study are only from residents of that region and if residents/ workers from other states were also included in the study. Pls clarify this in the methods.

4. Can the age of the participant who's models were included in the study affect the morphology. Please elaborate on this.

5. In the conclusion authors state that "The mandibular post-canine dentition exhibits significant variation in shape, which 33 can be attributed to the higher proportion of enamel in females and a higher proportion of dentin 34 in males."- this is a bold claim especially since this research did not involve any objectives to determine as such. Pls I would suggest to change/ remove this statement in conclusion. 

6. I suggest authors to include more references from the past 5 years for relevance as I have noted only 20 % references from past 5 years.

Author Response

Manuscript ID: applsci-2743216

Title: Geometric Morphometric shape analysis of Mandibular Post Canine Dentition

Authors: Srikant Natarajan, Junaid Ahmed *, Shravan Shetty, Nidhin Philip Jose, Sharada Chowdappa, Kavery Chengappa

Reviewer 1

I congratulate the authors for a well designed and executed research study. I would like to point out a few suggestions and improvements.

  1. Can the authors include the inclusion and exclusion criteria in a tabulated form with detailed criteriae.

We have included the inclusion and exclusion criterion in a tabular form and cited the same in the manuscript as shown

Line 123

Inclusion criterion

Exclusion Criterion

·       Individuals from Dakshinna Kannada between 13 - 20 years of age were included in the study

·       Any tooth in the posterior segment that has undergone restoration, or has fractures, caries, or crowns, is excluded from the study.

·       Patients exhibiting any developmental anomalies that affect the shape or size of the tooth, as well as individuals displaying symptoms of a syndrome, are excluded from the study.

·       Any tooth in the posterior segment that has undergone restoration, or has fractures, caries, or crowns, is excluded from the study.

·       Patients exhibiting any developmental anomalies that affect the shape or size of the tooth, as well as individuals displaying symptoms of a syndrome, are excluded from the study.

  1. How many operators were involved in the study and how did the authors take care of operator bias.

Line 150-158: We have included details regarding the number of observers involved in the study and reported the results of reliability analysis.

“Using a random subsample of 20 dental models, the assessment of intra- and interobserver error in landmark location was carried out. Over the course of five days, landmarks were marked independently by two observers. A single investigator obtained the landmarks to study for intra-observer variation. Reliability was measured using the Interclass Correlation Coefficient (ICC)[1], which ranged from 0.982 to 1.000, showing excellent reliability. Intra-operator reproducibility at the digitization stage was assessed using a Procrustes ANOVA. The average variance related to digitization error for centroid size as well as shape among the selected teeth was <10% leading to a conclusion that the errors were negligible compared to the total measurement effect. “

  1. Since this study was done in Dakshina Kannada region, how certain are the authors that the findings of the study are only from residents of that region and if residents/ workers from other states were also included in the study. Pls clarify this in the methods.

Line 115-121: We have provided further clarification on the inclusion of the study participants in the methodology section of our manuscript.

“Individuals were included in the study if they, along with their parents and grandparents, were born and raised in the Dakshina Kannada region. This three-generation residency criterion was established to ensure a strong regional connection among participants. Verification of this information was conducted through patient interviews at the dental clinic, where questions about the birthplace, residence, and occupation of their parents and grandparents were asked.

  1. Can the age of the participant who's models were included in the study affect the morphology. Please elaborate on this.

Line 108-113: We have expanded upon the selection criteria related to age and its potential confounding effects in the methodology section of our manuscript.

"After the age of 20, when the individual's occlusion has developed, the contacts and cusp tips tend to show attrition leading to the development of proximal contact areas and occlusal wear facets. This can result in incorrect landmark acquisition, particularly in terms of proximal contacts and cusp tips. To prevent such errors, our sample was drawn from the age range of 13–20 years, where the landmarks are stable and no physiological changes are observed so early in life.”

  1. In the conclusion authors state that "The mandibular post-canine dentition exhibits significant variation in shape, which 33 can be attributed to the higher proportion of enamel in females and a higher proportion of dentin 34 in males."- this is a bold claim especially since this research did not involve any objectives to determine as such. Pls I would suggest to change/ remove this statement in conclusion. 

Line 34-39: We have revised the conclusion in the abstract to align it more closely with the objectives of our study as suggested by you.

“This study shows how landmarks vary in permanent human post-canine dentition, a crucial finding for anatomic reconstruction and restorative dentistry. In particular, the molars and premolars of the mandible post-canine teeth are critical for achieving optimal masticatory efficiency and overall health.  Additionally, a higher degree of allometry and later formation of cusps correlate with greater shape variation, particularly on the distal and lingual sides. For precise restorative procedures, a thorough understanding of the anatomy of these teeth is required.”

  1. I suggest authors to include more references from the past 5 years for relevance as I have noted only 20 % references from past 5 years.

In accordance with the reviewers’ insightful suggestions, we have incorporated relevant and recent articles into our manuscript. Reference 17 has been added at Line 153, while References 28 to 31 have been included in the discussion section, specifically from Line 358 to 371. These added references are all recent publications, ranging from the years 2018 to 2023.

  1. López-Lázaro, S.; Alemán, I.; Viciano, J.; Irurita, J.; Botella, M.C. Sexual Dimorphism of the First Deciduous Molar: A Geometric Morphometric Approach. Forensic Sci. Int. 2018, 290, 94–102, doi:10.1016/j.forsciint.2018.06.036.
  2. Mahn, E.; Walls, S.; Jorquera, G.; Valdés, A.M.; Val, A.; Sampaio, C.S. Prevalence of Tooth Forms and Their Gender Correlation. J. Esthet. Restor. Dent. 2018, 30, 45–50, doi:10.1111/jerd.12341.
  3. Krenn, V.A.; Fornai, C.; Wurm, L.; Bookstein, F.L.; Haeusler, M.; Weber, G.W. Variation of 3D Outer and Inner Crown Morphology in Modern Human Mandibular Premolars. Am. J. Phys. Anthropol. 2019, 169, 646–663, doi:10.1002/ajpa.23858.
  4. Runte, C.; Dirksen, D. Symmetry and Aesthetics in Dentistry. Symmetry (Basel). 2021, 13, 1–19, doi:10.3390/sym13091741.
  5. Popovici, M.; Groza, V.; Petraru, O. Dental Morphological Variation in Chalcolithic and Bronze Age Human Populations from North-Eastern Romania. Ann. Anat. 2023, 245, 1–16, doi:10.1016/j.aanat.2022.152015.

Reviewer 2 Report

Comments and Suggestions for Authors

The article explores the impact of genetic and epigenetic alterations on the morphology of permanent mandibular premolars and molars using geometric morphometry. The utilization of 3D digital replicas of dental casts for analysis is commendable, as it allows for a detailed examination of tooth shape.

Firstly, the aim of the study is missing. That geometric morphometry is a powerful technique is not a sufficient scientific justification for an investigation, and certainly not for clinical implementation. 

The article lacks clarity regarding the selection criteria for the sample size of 160 dental casts. The rationale behind this specific number should be explicitly stated, and considerations for potential biases in the sample should be discussed to ensure the generalizability of the findings.

The methods employed, including procrustes superimposition, procrustes ANOVA, discriminant function analysis, and regression of shape over centroid size, are well-established techniques. Providing information on the reliability and reproducibility of the measurements would strengthen the methodological section.

Concerning the results I miss information on potential confounding variables that might influence the observed variations, such as age or ethnicity. 

The article should discuss potential alternative explanations for the observed variations and acknowledge the limitations of the study.  Moreover, the manuscript would benefit from a more nuanced discussion on the practical implications of the findings and how they could be translated into clinical practice.

Author Response

Manuscript ID: applsci-2743216

Title: Geometric Morphometric shape analysis of Mandibular Post Canine Dentition

Authors: Srikant Natarajan, Junaid Ahmed *, Shravan Shetty, Nidhin Philip Jose, Sharada Chowdappa, Kavery Chengappa

Reviewer 2

The article explores the impact of genetic and epigenetic alterations on the morphology of permanent mandibular premolars and molars using geometric morphometry. The utilization of 3D digital replicas of dental casts for analysis is commendable, as it allows for a detailed examination of tooth shape.

  1. Firstly, the aim of the study is missing. That geometric morphometry is a powerful technique is not a sufficient scientific justification for an investigation, and certainly not for clinical implementation. 

The aim has been added between Line 82 -86 as follows.

The aim of this study is to examine the change of geometric and anatomic landmarks, and thus the three-dimensional shape of the mandibular post-canine dentition, using 3D geometric morphometric analysis. Further, we aimed to determine the specific landmarks in post canine dentition that show sexual dimorphism

  1. The article lacks clarity regarding the selection criteria for the sample size of 160 dental casts. The rationale behind this specific number should be explicitly stated, and considerations for potential biases in the sample should be discussed to ensure the generalizability of the findings.

The selection criteria have been revised and now include both inclusion and exclusion criteria, which are presented in a table on line 123. We have also justified the sample size and adjusted the related statements accordingly. The inclusion of 120 samples has been justified, and we have provided a comprehensive explanation for the addition of 40 more samples, bringing the total to 160 samples in

Line 103 to 106

“The sample size determined was 120, but in order to account for the anatomic variation in the mandibular second premolar, an additional 40 casts were taken in order to examine the 60 male and 60 female casts of all post canine teeth.  

Line 144-147

“A total of 120 dental casts with the presence of all the 8 teeth, i.e. first premolar to second molar bilaterally, were selected for shape analysis.  These 120 casts consisted of 60 males and 60 females, and were specifically analyzed in relation to the mandibular first premolar, mandibular first molar, and mandibular second molar. In the sample of 120 patients, it was found that 37 had the two-cusp type of mandibular second premolar, while 83 had the three-cusp type. In order to increase the accuracy of our analysis, we expanded our sample size to include 40 additional casts (making the total sample number n=160)  This allowed us to examine the shape variation in both the two-cusp (n=49) and three-cusp (n=111) types of mandibular second premolars. “

  1. The methods employed, including procrustes superimposition, procrustes ANOVA, discriminant function analysis, and regression of shape over centroid size, are well-established techniques. Providing information on the reliability and reproducibility of the measurements would strengthen the methodological section.

Line 150-158: The following details have been added about the digitization errors as well as the landmarks' reliability and reproducibility:

“Using a random subsample of 20 dental models, the assessment of intra- and interobserver error in landmark location was carried out. Over the course of five days, landmarks were marked independently by two observers. A single investigator obtained the landmarks to study for intra-observer variation. Reliability was measured using the Interclass Correlation Coefficient (ICC)[1], which ranged from 0.982 to 1.000, showing excellent reliability. Intra-operator reproducibility at the digitization stage was assessed using a Procrustes ANOVA. The average variance related to digitization error for centroid size as well as shape among the selected teeth was <10% leading to a conclusion that the errors were negligible compared to the total measurement effect.”

  1. Concerning the results I miss information on potential confounding variables that might influence the observed variations, such as age or ethnicity. 

The formation of the tooth takes place intraosseous, and the shape of which is determined genetically. The shape changes with age and once it erupts the shape is fixed as the cells forming the enamel are no longer available for modulating the shape of enamel. Thus, once erupted, age is not a confounding factor for the shape of the teeth. In terms of ethnicity, our inclusion criterion was population of Dakshina Kannada which we have included in the methodology as follows in line 115-120.

“Individuals were included in the study if they, along with their parents and grandparents, were born and raised in the Dakshina Kannada region. This three-generation residency criterion was established to ensure a strong regional connection among participants. Verification of this information was conducted through patient interviews at the dental clinic, where questions about the birthplace, residence, and occupation of their parents and grandparents were asked.”

  1. The article should discuss potential alternative explanations for the observed variations and acknowledge the limitations of the study.  Moreover, the manuscript would benefit from a more nuanced discussion on the practical implications of the findings and how they could be translated into clinical practice.

Line 358-371: We have updated the manuscript to include five contemporary and pertinent papers. These articles delve into the forensic, functional efficiency, anthropological, and clinical dentistry aspects of the current study

“Mahn E et al (2018) correlated the anterior tooth form and gender and found no significant difference between them. They proposed 5 new hybrid tooth forms which can be prefabricated for patient use in esthetics.  Using our study of the posterior tooth landmarks clinically and functionally superior preformed crowns can be fabri-cated specifically for premolars and molars.

Similar to the study by Krenn V et al (2019) we found that the mandibular premolars vary more on the lingual side than the buccal segment. They also found that the two mandibular premolars covary with each other.  Thus the landmark data of available teeth, can be further assessed to predict the shape of missing teeth which can be useful to reproduce the crown clinically.

Symmetry in the dental arches are clinically relevant. The presence of directional asymmetry i.e. consistently larger or smaller teeth on one side may be useful feature for the practice of orthodontics and alignment.

Understanding the landmarks can also be useful for anthropological and forensic studies. The landmark changes from early humans to modern humans can give good insights into the evolution of tooth shape.”

Round 2

Reviewer 2 Report

Comments and Suggestions for Authors

All comments are answered.

Author Response

Dear Editor,

Thank you for the insightful suggestions. We have made the necessary corrections accordingly, and a point-by-point response is included below. The corrections in the manuscript have been highlighted in green for your review.

regards,

Dr. Junaid and Dr. Srikant

Key words. “dentistry” and “tooth” could be added in my opinion.
Line 41: The key words “Dentistry” and “Tooth” have been included as per the editors suggestions

Introduction. Authors stated “Genetics, environment, and epigenetics all play a role in determining the shape of an animal's teeth”. Please add a reference for this statement.

Line 45: The reference for this statement is added as reference 1

Introduction. Authors stated “For example, it can help us better understand the evolution of different species and how they are related to one another”. Please remove “For example”.
Line 49: The statement has been modified and the “for example”, is removed

Materials and Methods. Authors stated “Prospective pretreatment dental casts (n=160) were obtained from the archives of the Department of Orthodontics at Manipal College of Dental Sciences, Mangalore, as per the sample size calculation mentioned above”. Please add if and how sample size calculation has been performed.
Line 88-100: Sample size calculation is added in the beginning of the materials and methods.

Materials and Methods. Please add a reference for each method.

Reference 14 (line 131), Reference 5 and 15 ( line 133), Reference 16 in line 139,  represent the methodology are added

Materials and Methods. For each material used, please add details about commercial name manufacturer, City and State.
Line 125-128: In our study two materials were used, one for impressions and another pouring the cast, for both the details of the commercial name and manufacturer details are added. 

After obtaining the impressions (taken using Dentsply Aquasil Soft Putty And Kit, Dentsply, India) and pouring the casts using Type 4 Extra Hard Dental Die Stone (Zhermack Badia Polesine (RO), Italy)

Materials and Methods. For each machinery used, please add details about commercial name manufacturer, City and State.
Line 127-128: The lab scanner was used for the study the manufacturer name and country is included

 inEOS X5-Lab scanner (Dentsply Sirona, India).

Materials and Methods. Please add details about software used, version, Manufacturer, City and State.
The software, manufacturer details with year is mentioned as follows

Line 139:  3D Slicer software (http://www.slicer.org).

Line 159-162:  “The landmark coordinates were tabulated and processed in Microsoft Excel (Microsoft Corporation, 2023) and imported to the morphometry software MorphoJ (version 1.08.01, for Microsoft Windows, Klingenberg, C. P. 2011)”

Materials and Methods. Authors stated “Intra-operator reproducibility at the digitization stage was assessed using a Procrustes ANOVA” ANOVA is used for gaussian distributions. Please explain how normality of data was tested.

The analysis of 3D landmark data involves the use of Procrustes ANOVA, supplemented with randomized permutation bootstrapping. This approach represents a non-parametric variant of the conventional analysis

Line 161-162: The analysis was done using one–way nonparametric ANOVA with randomized permutation procedure (10,000 iterations) which is a

Materials and Methods. Statistics. Please add significance level for P values (0.05? 0.01?).
Please add statistical software details.

Line 176-177: We have added a statement for level of significance taken for the study.

“A 95% confidence interval was used to determine the level of significance. A p-value of less than or equal to 0.05 was considered statistically significant.”

The details of the two softwares used are mentioned in the following lines

Line 159:  (IBM, SPSS version 20.0, Chicago USA)

Line 167-168: MorphoJ (version 1.08.01, for Microsoft Windows, Klingenberg, C. P. 2011, https://morphometrics.uk/MorphoJ_page.html)

Results. Please add P values all along this section.

Line 197 to Line 228: p values for the significant and non-significant variables between the sexes using procrustes ANOVA as well as discriminant function analysis is entered and highlighted.

Discussion. Authors stated “The landmark changes from early humans to modern humans can give good insights into the evolution of tooth shape. [17,31]”. In fact the present report tested casts of patients in the age range of 13-20. Provide a general interpretation of the results in the context of other evidence, and implications for future research. It could be added that “Moreover, the present report tested casts of patients in the age range of 13-20. It would be interesting in the future to test also other ages, taking into careful account also the effects of remineralizing agents such as fluoride (Historical and bibliometric notes on the use of fluoride in caries prevention. Zampetti P, Scribante A. Eur J Paediatr Dent. 2020 Jun;21(2):148-152.), casein phosphopeptide-amorphous calcium phosphate (Efficacy of a Novel Bioactive Glass-Polymer Composite for Enamel Remineralization following Erosive Challenge. Fallahzadeh F, Heidari S, Najafi F, Hajihasani M, Noshiri N, Nazari NF. Int J Dent. 2022 Apr 22;2022:6539671. doi: 10.1155/2022/6539671..) and biomimetic hydroxiapathite (Biomimetic hydroxyapatite paste for molar-incisor hypomineralization: A randomized clinical trial. Butera A, Pascadopoli M, Pellegrini M, Trapani B, Gallo S, Radu M, et al. Oral Dis. 2022 Sep 22. doi: 10.1111/odi.14388.) in order to evaluate the interaction between age and preventive treatments”.

Line 395-399: Thank you for the insightful suggestion on adding future perspectives on how preventive treatment procedures could alter tooth anatomy. We have incorporated these considerations as shown below.

 “Moreover, the present report tested casts of patients in the age range of 13-20. It would be interesting in the future to test also other ages, taking into careful account also the effects of remineralizing agents such as fluoride [33], casein phosphopep-tide-amorphous calcium phosphate [34] and biomimetic hydroxyapatite [35] in order to evaluate the interaction between age and preventive treatments.

These concerns should be added to Discussion section.
Discussion. Please elongate a bit this section.
Line 371 to 385 and lines 395-399 are added to the discussion that add content to the discussion

Discussion. Please add a paragraph concerning the treatment alternatives.

Line 371 to 385 and lines 395-399 are added to the discussion that discuss about the treatment alternatives

Discussion. Please add a paragraph showing the limitations of the present report.

Line 400  to 407: We have added the limitations of the study

Figure 1. Please enlarge the figure in order to increase readability.
Figure 2. Please enlarge the figure in order to increase readability.

We have enlarged the figures as suggested to improve the readability

Tables. Please add some tables showing the results and descriptive statistics of the main variables tested.

Line 153: We have added a new table on demographic details, sex distribution and landmark related data as table 1. The same has been cited in line 152 and 181.

Parameter

Details

Age

Total  (Mean ± Standard Deviation)

18.62 ± 2.49

Female (Mean ± Standard Deviation)

17.85 ± 2.81

Male(Mean ± Standard Deviation)

19.47 ± 1.74

Sex

Male (n,%)

76 (47.5%)

Female (n,%)

84 (52.5%)

Distribution of cases

Tooth

Male

Female

Mandibular First Premolar

60

60

Mandibular Second Premolar (two Cusp Type)

29

20

Mandibular Second Premolar (Three Cusp Type)

47

64

Mandibular First Molar

60

60

Mandibular Second Molar

60

60

Number of Landmarks

Tooth

Geometric Evidence

Anatomic Evidence

Mandibular First Premolar

11

9

Mandibular Second Premolar (two Cusp Type)

11

8

Mandibular Second Premolar (Three Cusp Type)

13

8

Mandibular First Molar

19

13

Mandibular Second Molar

15

12
